# Correctness-Optimized Residual Activation Lens (CORAL): Transferable and Calibration-Aware Inference-Time Steering

**Miranda Muqing Miao** [1]  **Young-Min Cho** [1]  **Lyle Ungar** [1]

## Abstract

Large language models (LLMs) exhibit persistent miscalibration, especially after instruction tuning and preference alignment. Modified training objectives can improve calibration, but retraining is expensive. Inference-time steering offers a lightweight alternative, yet most existing methods optimize proxies for correctness rather than correctness itself. We introduce CORAL (Correctness-Optimized Residual Activation Lens), a regularized inference-time steering method that captures distributed correctness signals from model internal activations using weight-decay MLP probes. We evaluate CORAL across three 7B-parameter models and find that it consistently improves accuracy by 10% and expected calibration error (ECE) by 50% on average. We additionally demonstrate that these gains transfer without retraining to the complete published test sets of four held-out benchmarks (ARC-Challenge, HellaSwag, Math-MC, OpenBookQA), averaging 14% accuracy improvements and 49% ECE improvements. Our results support the hypothesis that distributed information in model internals can be extracted using regularized probes when individual neurons are insufficient. CORAL thus provides a compute-efficient, transferable, and calibration-aware approach to improve MCQA performance during inference.

## 1. Introduction

Inference-time steering has emerged as a promising approach for modifying LLM behavior without retraining overhead. Most steering methods target behavioral goals such as honesty, refusal, or stylistic attributes (Rimsky et al., 2024). In this paper, we focus on a more direct objective: steering for factuality while simultaneously improving calibration. This dual goal is important because miscalibration persists across models and post-training pipelines, and common alignment methods like Reinforcement Learning with Human Feedback (RLHF) and Direct Policy Optimization (DPO) worsen calibration even as they improve other metrics (Jiang et al., 2021; Leng et al., 2025; Ouyang et al., 2022; Rafailov et al., 2023).

Steering methods divide into two paradigms. Sparse approaches search for discrete, localized features, often using Sparse Autoencoders (SAEs) to identify individual features that causally influence behavior (Bricken et al., 2023; Templeton et al., 2024). Distributed approaches instead treat the relevant signal as spread across many dimensions (Zou et al., 2025).

Recent mechanistic work suggests that confidence and correctness signals fall into the distributed category. While some behaviors localize to interpretable circuits, many others cannot be fully captured by individual features (Olah et al., 2024; Lindsey et al., 2025). We focus on the distributed setting and show that regularized probes extract correctness signals that sparse methods miss.

Within distributed steering for factuality and calibration, three recent methods are most relevant: Inference-Time Intervention (ITI) (Li et al., 2023), SteerConf (Zhou et al., 2025), and Calibrating LLM Confidence by Probing Perturbed Representation Stability (CCPS) (Khanmohammadi et al., 2025). ITI performs distributed steering on attention head activations to improve truthfulness, but its effect on factual correctness remains underexplored. SteerConf elicits verbalized confidence through prompting and uses expressed confidence as a proxy for calibration. CCPS measures representation stability under adversarial perturbations to predict correctness, achieving calibration improvements and occasional accuracy gains. These methods share two limitations: they optimize proxies for correctness rather than correctness itself, and they do not leverage internal activations to directly steer model outputs. Furthermore, none of them demonstrates transferability to out-of-distribution benchmarks.

---

[1]Department of Computer and Information Science, University of Pennsylvania, Philadelphia, USA. Correspondence to: Miranda Muqing Miao <miaom@seas.upenn.edu>.

*Proceedings of the $43^{rd}$ International Conference on Machine Learning*, Seoul, South Korea. PMLR 306, 2026. Copyright 2026 by the author(s).

In this paper, we address these gaps with CORAL, a regularized inference-time steering method that captures distributed correctness signals from model internal activations using weight-decay MLP probes. We train a regularized probe that predicts correctness directly from frozen residual stream activations, then use these predictions to steer model behavior at inference time. By jointly optimizing for the accuracy and calibration components of the Brier score, CORAL improves both metrics and transfers its impact successfully to unseen benchmarks without any retraining.

Our contributions are as follows:

1. We introduce CORAL, a lightweight weight-decay MLP probe that consistently improves both accuracy and calibration across three 7B-parameter models and multiple MCQA benchmarks, using only 8.4k–10k training questions.

2. We demonstrate that CORAL transfers without retraining to the complete published test sets of four held-out benchmarks (ARC-Challenge, HellaSwag, Math-MC, OpenBookQA), averaging 14% accuracy improvements and 49% ECE improvements. This transfer indicates that the probe captures a general-purpose correctness subspace rather than task-specific patterns.

3. We provide evidence that correctness signals are distributed across many features. SAE-based ablations show negligible individual neural effects, while regularized probes successfully aggregate these distributed signals into effective steering vectors.

## 2. Related Works

**Probing and internal representations.** Linear probes have long been used to decode task-relevant variables from intermediate representations (Alain & Bengio, 2017). Research on transformer feed-forward layers shows that residual streams contain rich, interpretable information that accumulates across layers (Geva et al., 2021; 2022). Recent studies demonstrate that LLM activations can predict answer correctness from question-only signals, though this work focuses on prediction rather than intervention (Cencerrado et al., 2025).

**Inference-time intervention (ITI).** ITI methods modify activations during forward passes to shift model behavior, and attention-head interventions have improved truthfulness by targeting a small set of heads (Li et al., 2023). Contrastive activation addition instead trains steering vectors from positive and negative examples to control behavioral attributes (Rimsky et al., 2024). These methods demonstrate that activation edits can causally influence outputs, but they typically target softer behavioral goals rather than factuality directly.

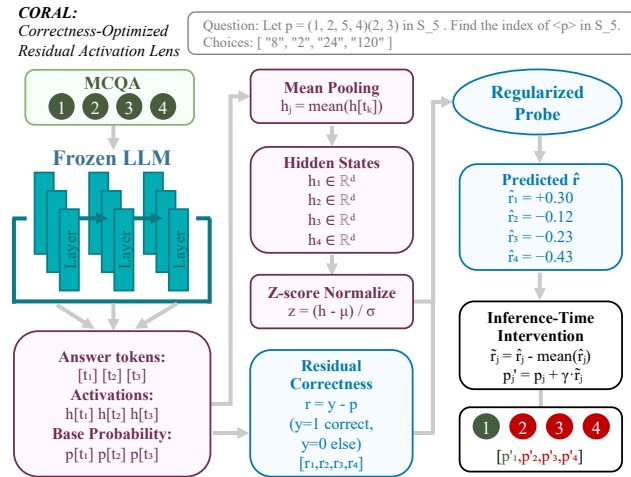

*Figure 1.* Overview of CORAL. Given an MCQA question, a frozen LLM produces per-option hidden states and base probabilities. Hidden states are mean-pooled over answer tokens, z-score normalized, and passed through a trained probe to predict residual correctness. At inference, predicted residuals are centered and used to steer base probabilities toward better calibration.

**Sparse Autoencoders and mechanistic interpretability.** SAEs decompose activations into sparse, approximately monosemantic features (Bricken et al., 2023; Huben et al., 2024). Practitioners use SAE features for steering or ablation because they appear more interpretable than raw neurons (Templeton et al., 2024; Gao et al., 2024). However, circuit tracing work reveals that many model behaviors arise from distributed computation across many features rather than isolated units (Conmy et al., 2023; Ameisen et al., 2025; Lindsey et al., 2025). Our SAE ablation experiments confirm this finding for correctness signals: individual features produce negligible causal effects.

**Calibration in LLMs.** Neural networks, including LLMs, often exhibit miscalibration where predicted confidence fails to match empirical accuracy (Guo et al., 2017; Jiang et al., 2021). Post-hoc methods like temperature scaling and isotonic regression adjust output probabilities but cannot improve accuracy (Guo et al., 2017; Zadrozny & Elkan, 2002). Recent work identifies entropy neurons that regulate confidence without affecting task performance (Stolfo et al., 2024), and shows that calibration information evolves across layers (Joshi et al., 2025). CORAL differs from these approaches by using internal activations to jointly improve accuracy and calibration through direct residual prediction.

## 3. Methodology

Having established that existing methods optimize proxies rather than correctness directly, we now describe CORAL's methodology. Probing literature shows that intermediate representations contain rich task-relevant information beyond what surfaces in final predictions. We exploit this

insight by training lightweight probes on frozen activations to predict residual correctness: the gap between a model's assigned probability and the ideal target. This formulation directly optimizes the Brier score and enables inference-time steering that improves both accuracy and calibration without modifying model weights.

We first formalize residual correctness (3.1), then describe CORAL's architecture (3.2), and finally detail our experimental setup (3.3).

### 3.1. Problem Formulation: Residual Correctness

A model is well-calibrated when its stated confidence matches its empirical accuracy. Calibration means that among questions where the model assigns 90% confidence, approximately 90% should be answered correctly. When this alignment breaks, confidence scores lose their decision-theoretic value.

Consider a multiple-choice question answering task where a model assigns probability $p_j$ to each option $j \in \{1, 2, \ldots, n\}$. We define residual correctness $r_j$ as the gap between the ideal target distribution and the model's prediction:

$$r_j = \begin{cases} 1 - p_j & \text{if option } j \text{ is correct} \\ -p_j & \text{if option } j \text{ is incorrect} \end{cases} \quad (1)$$

Positive residuals indicate underconfidence on correct answers; negative residuals indicate overconfidence on incorrect answers. A perfect confidence predictor assigning probability 1 to the correct option and 0 to all others would have $r_j = 0$ for all options.

This formulation directly targets the Brier score, a proper scoring rule. The Brier score decomposes as $\sum_j (p_j - y_j)^2 = \sum_j r_j^2$, where $y_j \in \{0, 1\}$ indicates correctness. Training a probe to predict $\hat{r}_j$ and applying the correction $p_j' = p_j + \hat{r}_j$ minimizes expected squared error between predictions and ground truth. CORAL thus optimizes a principled calibration objective rather than fitting an empirical pattern.

### 3.2. CORAL Architecture

CORAL consists of three components: activation extraction, a regularized MLP probe, and inference-time steering.

#### 3.2.1. ACTIVATION EXTRACTION

For each question with $n$ answer options, we perform $n$ forward passes, concatenating the question prompt with each candidate answer. We record hidden states $\mathbf{h}_{j,t}^{(l)} \in \mathbb{R}^d$ at layer $l$ while the model processes answer tokens, then mean-pool over token positions:

$$\mathbf{h}_j^{(l)} = \frac{1}{T_j} \sum_{t=T_{\text{prompt}}+1}^{T_{\text{prompt}}+T_j} \mathbf{h}_{j,t}^{(l)} \quad (2)$$

The resulting dataset contains $N \times n$ activation vectors, where $N$ is the number of questions. Each vector has dimensionality $d_{\text{model}}$ (e.g., 4096 for 7B parameter models). We pair each activation with its corresponding residual correctness label $r_j$, computed from the model's softmax probabilities and ground truth. This structure preserves the grouped nature of the data, where each question contributes exactly $n$ samples corresponding to its answer options. In the end, we apply z-score normalization using training set statistics before passing activations to the probe.

#### 3.2.2. WEIGHT-DECAY MLP PROBE

We train a four-hidden-layer MLP (dimensions 1024, 512, 256, 128) with ReLU activations, dropout (p=0.2), and tanh output to bound predictions to $[-1, 1]$. The loss combines mean squared error with an output penalty:

$$\mathcal{L} = \frac{1}{N} \sum_{i=1}^{N} (r_i - \hat{r}_i)^2 + \lambda_{\text{out}} \cdot \frac{1}{N} \sum_{i=1}^{N} \hat{r}_i^2 \quad (3)$$

We optimize using AdamW with weight decay, selecting hyperparameters via grid search over validation $R^2$. Training requires under 5 hours on a single GPU.

#### 3.2.3. INFERENCE-TIME STEERING

At inference, we extract activations for each answer option, compute probe predictions $\hat{r}_j$, center them to ensure zero-sum, and apply additive corrections:

$$p_j' = \frac{\max(p_j + \gamma \cdot \tilde{r}_j, 0)}{\sum_{j'=1}^{n} \max(p_{j'} + \gamma \cdot \tilde{r}_{j'}, 0)} \quad (4)$$

where $\tilde{r}_j = \hat{r}_j - \frac{1}{n} \sum_{j'} \hat{r}_{j'}$ and $\gamma$ controls steering strength. This shifts probability mass toward options the probe identifies as correct while maintaining a valid distribution.

### 3.3. Experimental Setup

#### 3.3.1. TRAINING DATASETS

We train two independent probes on two sets of datasets to demonstrate the generalization and robustness of CORAL.

**Construct Training Dataset.** We first construct a custom training dataset by aggregating questions from two reasoning benchmarks. Specifically, we evenly sample 5,000 questions each from CommonsenseQA and RACE **training** questions, yielding 10,000 total training questions (Talmor et al., 2019; Lai et al., 2017). We partition this construct

dataset into 80% train / 20% validation at the question level. We use GroupKFold to ensure all options from the same question remain in the same split, preventing information leakage across partitions. We evaluate on those benchmarks' corresponding and complete **test** questions. We call the probe trained on this Construct Training Dataset **Probe 1.**

**MMLU Split Dataset.** We also train a probe based on Massive Multitask Language Understanding (MMLU), which is a benchmark that tests the models' knowledge and reasoning on a broad range of topics, using multiple choice questions (Hendrycks et al., 2021). We partition the 14k test questions from MMLU into training (8,400 questions, 60%), validation (2,800 questions, 20%), and evaluation (2,800 questions, 20%) splits. This is because MMLU provides no in-distribution training split. The dev set contains only 1,531 questions, which is insufficient for training probes over 4,096-dimensional activations. The 90k auxiliary train dataset of MMLU contains a vast number of questions included in other benchmarks. Training on the 90k auxiliary dataset would therefore contaminate our transfer experiments. We call the probe trained on this MMLU Split Dataset **Probe 2.**

**Prompting.** We collect activations strictly following few-shot prompting template from EleutherAI `lm-eval` harness templates (Sutawika et al., 2023). For each question, we extract mean-pooled hidden states over answer tokens at each layer.

**Validation.** We solely use the validation data sets to determine the optimal steering layer and steering strength $\gamma$.

### 3.3.2. TRANSFER BENCHMARKS

We evaluate generalization on the complete published test sets of four held-out benchmarks: HellaSwag, OpenBookQA, Math-MC, and ARC-Challenge (Li et al., 2025; Banerjee et al., 2019; Amini et al., 2019; Clark et al., 2018). These benchmarks share no questions with training data and evaluate CORAL across tasks spanning commonsense NLP completion, multi-step question answering, math reasoning, and abstraction reasoning.

### 3.3.3. EVALUATION METRICS

We report accuracy and four calibration metrics: Expected Calibration Error (ECE), class-wise ECE (cwECE), Brier score, and negative log-likelihood (NLL). ECE measures alignment between confidence and accuracy across binned predictions. The Brier score captures both accuracy and calibration as mean squared error against ground truth. Lower values indicate better performance for all calibration metrics.

**Accuracy.** For each question, we select the answer option with the highest probability and compare it to the ground truth label:

$$\text{Accuracy} = \frac{1}{N} \sum_{i=1}^{N} \mathbf{1} \left[ \arg\max_j p_{i,j} = y_i \right] \qquad (5)$$

where $N$ is the number of questions, $p_{i,j}$ is the probability assigned to option $j$ for question $i$, and $y_i$ is the correct answer.

**Expected Calibration Error (ECE).** ECE measures the alignment between predicted confidence and actual accuracy. We partition predictions into $B$ equally-spaced bins based on the maximum probability (confidence) and compute the weighted average of the gap between accuracy and confidence within each bin:

$$\text{ECE} = \sum_{b=1}^{B} \frac{|S_b|}{N} \left| \text{acc}(S_b) - \text{conf}(S_b) \right| \qquad (6)$$

where $S_b$ is the set of samples whose confidence falls in bin $b$, $\text{acc}(S_b)$ is the accuracy of samples in bin $b$, and $\text{conf}(S_b)$ is the average confidence in bin $b$. Lower ECE indicates better calibration. A perfectly calibrated model should have ECE = 0. For all statistics reported, the chosen bin count $B$ = 25 unless otherwise stated.

**Class-wise Expected Calibration Error (cwECE).** While ECE only considers the top predicted class, cwECE evaluates calibration across all answer options. For each class $c \in \{1, \dots, n\}$, we compute a per-class ECE by binning the predicted probabilities $p_{i,c}$ and comparing them to the binary correctness indicator $\mathbf{1}[y_i = c]$:

$$\text{ECE}_c = \sum_{b=1}^{B} \frac{|S_{b,c}|}{N} \left| \text{acc}_c(S_{b,c}) - \text{conf}_c(S_{b,c}) \right| \qquad (7)$$

where $S_{b,c}$ contains samples whose probability for class $c$ falls in bin $b$, $\text{acc}_c(S_{b,c})$ is the fraction of samples in the bin where class $c$ is correct, and $\text{conf}_c(S_{b,c})$ is the average probability for class $c$ in the bin. The overall cwECE is the average across all classes:

$$\text{cwECE} = \frac{1}{n} \sum_{c=1}^{n} \text{ECE}_c \qquad (8)$$

For all statistics reported, the chosen bin count $B$ = 25 unless otherwise stated.

**Brier Score.** The Brier score measures the mean squared error between predicted probabilities and ground truth labels. For each question $i$ with predicted distribution

$\mathbf{p}_i = (p_{i,1}, \ldots, p_{i,n})$ and correct answer $y_i$, the Brier score is:

$$\text{Brier} = \frac{1}{N} \sum_{i=1}^{N} \sum_{j=1}^{n} (p_{i,j} - \mathbb{1}[j = y_i])^2 \qquad (9)$$

where $\mathbb{1}[j = y_i]$ equals 1 if option $j$ is correct and 0 otherwise. Lower Brier scores indicate better calibrated and more accurate predictions. Unlike ECE, the Brier score is a proper scoring rule that jointly rewards accuracy and calibration.

**Negative Log-Likelihood (NLL).** We report a binary cross-entropy measure on top-1 confidence. Let $p_{i,\hat{y}_i} = \max_j p_{i,j}$ denote the model's maximum probability for question $i$, and let $c_i \in \{0, 1\}$ indicate whether the top-1 prediction is correct. We compute:

$$\text{NLL} = -\frac{100}{N} \sum_{i=1}^{N} \left[ c_i \log p_{i,\hat{y}_i} + (1 - c_i) \log(1 - p_{i,\hat{y}_i}) \right] \qquad (10)$$

where the factor of 100 rescales for readability. This formulation provides a consistent measure across benchmarks with different numbers of answer options (3, 4, or 5), because it evaluates confidence in the selected answer rather than across all classes. Note that the standard multiclass NLL bound of $\ln(K)$ does not apply to this binary formulation.

### 3.3.4. BASELINES

We compare CORAL against established steering and calibration methods. These include (1) Inference Time Intervention (ITI) (Li et al., 2023), which performs activation steering at attention heads rather than residual stream positions; (2) SteerConf (Zhou et al., 2025), which targets confidence elicitation as a proxy for correctness; (3) Contrastive Confidence Probing and Steering (CCPS) (Khanmohammadi et al., 2025), which uses contrastive learning on correct versus incorrect activations. We also report Few-Shot Prompting results using the Eval Harness framework (Sutawika et al., 2023). Detailed descriptions of baseline implementations are provided in Appendix B.

## 4. CORAL Results

In this section, we present CORAL's impact on in-distribution test questions, a layer analysis, and CORAL's success on out-of-distribution transfer benchmarks.

### 4.1. CORAL Steering Results

CORAL steering produces consistent gains in both accuracy and calibration across all three models. Table 1 reports single-layer results for each model, with optimal layers selected via validation only.

We sweep steering strength $\gamma \in \{0.25, 0.5, \ldots, 2.75, 3.0\}$ on the validation set. The parameter $\gamma$ controls the magni-

tude of activation shifts along the learned correctness direction. We find that $\gamma = 1$ produces the strongest and most consistent results, suggesting that unit-magnitude shifts align well with the natural scale of residual stream activations.

CORAL yields leading performance across accuracy and most calibration metrics. On RACE, CORAL outperforms the second-best method by 17.39 percentage points in accuracy and achieves a 64% relative improvement in ECE using DeepSeek-LLM-7B-Chat. On CommonsenseQA and MMLU, CORAL consistently achieves the lowest or second lowest Brier scores across all three model families while maintaining meaningfully competitive accuracy.

Probe architecture has minimal impact on performance. MLPs with one to three hidden layers underperform linear probes by 0.5%–0.75%, while deeper architectures yield no further gains. This pattern indicates that extracting residual correctness signals requires only modest nonlinearity.

We also investigated multi-layer CORAL, where probes are trained on concatenated activations from the top 3-5 layers. The motivation was to capture correctness signals that might be distributed across the model's later processing stages. However, this approach produced no meaningful improvement over single-layer steering. This suggests that correctness-relevant information is sufficiently concentrated at individual layers, and aggregating across layers does not provide additional meaningful signals.

### 4.2. Layer Analysis

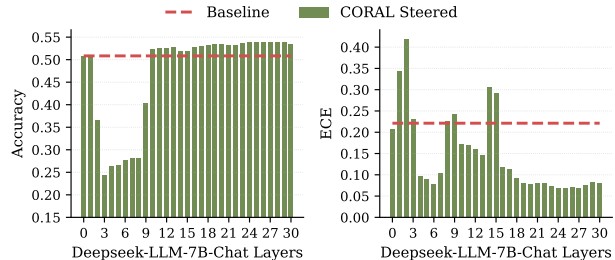

*Figure 2.* Layer-wise steering performance for DeepSeek-7B-Chat. Left: accuracy across layers 0–30 for `lm-eval` harness baseline (dotted red lines) and CORAL performance (green bars). Right: corresponding ECE values.

Figure 2 examines which layers contain the most useful correctness signals for steering. We train separate probes at each layer of DeepSeek-7B-Chat and evaluate steering performance independently.

Middle layers contain the most useful correctness signals for steering. Layers 17–21 yield the strongest improvements in both accuracy and calibration, while early and late layers provide noisy results. At early layers (0–6), steering produces accuracy below the unsteered baseline. Performance improves steadily through layers 9–15, peaks in the middle

*Table 1.* Comparison of steering methods across different models and benchmarks. Blue rows are CORAL-steered results. **Bold** numbers indicate best performance and underlined numbers indicate second best under Accuracy, ECE, cwECE, NLL, and Brier Score. Baseline methods are SteerConf (Zhou et al., 2025), Few-Shot Prompting (Eval Harness) (Sutawika et al., 2023), Inference Time Intervention (ITI) (Li et al., 2023), and CCPS (Zadrozny & Elkan, 2002; Guo et al., 2017; Khanmohammadi et al., 2025).

### MMLU Test Subset

| Mistral-7B-Instruct-v0.3 | | | | | | Qwen2.5-7B-Instruct | | | | | | DeepSeek-7B-Chat | | | | |
|---|---|---|---|---|---|---|---|---|---|---|---|---|---|---|---|---|
| Method | Acc ↑ | ECE ↓ | cwECE ↓ | NLL ↓ | Brier ↓ | Method | Acc ↑ | ECE ↓ | cwECE ↓ | NLL ↓ | Brier ↓ | Method | Acc ↑ | ECE ↓ | cwECE ↓ | NLL ↓ | Brier ↓ |
| SteerConf | 60.53 | 15.47 | 7.16 | 146.54 | 20.66 | SteerConf | 63.94 | 15.67 | 8.75 | 113.96 | 20.57 | SteerConf | 37.48 | 23.39 | 13.89 | 91.15 | 30.46 |
| Few-Shot | 61.34 | 18.46 | 9.79 | 72.44 | 22.62 | Few-Shot | 73.19 | 16.36 | 8.83 | 73.98 | 19.67 | Few-Shot | 50.84 | 22.12 | 11.44 | 81.86 | 26.27 |
| ITI | 59.02 | 17.74 | 10.09 | 69.61 | 22.63 | ITI | 70.42 | 15.72 | 9.10 | 71.09 | 19.68 | ITI | 48.91 | 21.26 | 11.78 | 78.66 | 26.28 |
| CCPS | 61.20 | 8.36 | 10.07 | 55.49 | 18.95 | CCPS | 73.02 | 7.41 | 9.09 | 56.67 | 16.48 | CCPS | 51.10 | 8.38 | 12.05 | 65.86 | 22.60 |
| CORAL | 64.22 | 3.27 | 2.83 | 54.30 | 17.92 | CORAL | 74.23 | 4.23 | 2.77 | 46.61 | 15.51 | CORAL | 53.97 | 6.21 | 4.36 | 61.95 | 21.33 |

### RACE

| Mistral-7B-Instruct-v0.3 | | | | | | Qwen2.5-7B-Instruct | | | | | | DeepSeek-7B-Chat | | | | |
|---|---|---|---|---|---|---|---|---|---|---|---|---|---|---|---|---|
| Method | Acc ↑ | ECE ↓ | cwECE ↓ | NLL ↓ | Brier ↓ | Method | Acc ↑ | ECE ↓ | cwECE ↓ | NLL ↓ | Brier ↓ | Method | Acc ↑ | ECE ↓ | cwECE ↓ | NLL ↓ | Brier ↓ |
| SteerConf | 40.83 | 4.67 | 4.78 | 122.27 | 26.52 | SteerConf | 45.69 | 21.45 | 10.87 | 104.68 | 27.11 | SteerConf | 50.83 | 10.66 | 8.52 | 70.15 | 25.10 |
| Few-Shot | 61.43 | 3.18 | 3.72 | 57.59 | 19.72 | Few-Shot | 58.37 | 10.60 | 6.44 | 64.84 | 21.48 | Few-Shot | 55.67 | 4.40 | 4.16 | 61.59 | 21.18 |
| ITI | 61.45 | 11.97 | 7.54 | 68.43 | 17.77 | ITI | 58.34 | 11.46 | 13.06 | 70.45 | 19.36 | ITI | 55.63 | 16.56 | 8.44 | 73.18 | 19.09 |
| CCPS | 60.56 | 2.45 | 3.29 | 57.80 | 19.77 | CCPS | 58.38 | 5.11 | 6.44 | 55.22 | 15.55 | CCPS | 54.90 | 3.00 | 4.25 | 60.36 | 20.84 |
| CORAL | 74.93 | 3.08 | 3.18 | 55.06 | 15.63 | CORAL | 71.04 | 3.46 | 3.35 | 54.40 | 16.88 | CORAL | 73.06 | 2.62 | 3.08 | 57.69 | 16.06 |

### CommonsenseQA

| Mistral-7B-Instruct-v0.3 | | | | | | Qwen2.5-7B-Instruct | | | | | | DeepSeek-7B-Chat | | | | |
|---|---|---|---|---|---|---|---|---|---|---|---|---|---|---|---|---|
| Method | Acc ↑ | ECE ↓ | cwECE ↓ | NLL ↓ | Brier ↓ | Method | Acc ↑ | ECE ↓ | cwECE ↓ | NLL ↓ | Brier ↓ | Method | Acc ↑ | ECE ↓ | cwECE ↓ | NLL ↓ | Brier ↓ |
| SteerConf | 29.24 | 33.71 | 14.09 | 142.01 | 34.56 | SteerConf | 39.81 | 18.54 | 7.77 | 134.80 | 20.16 | SteerConf | 38.57 | 13.12 | 6.64 | 69.41 | 24.72 |
| Few-Shot | 73.71 | 15.00 | 6.74 | 63.47 | 18.44 | Few-Shot | 84.68 | 12.16 | 5.25 | 83.59 | 12.94 | Few-Shot | 70.52 | 15.71 | 7.27 | 67.63 | 19.61 |
| ITI | 66.69 | 16.29 | 7.67 | 68.36 | 21.74 | ITI | 78.61 | 13.21 | 5.97 | 90.03 | 15.26 | ITI | 63.80 | 17.06 | 8.27 | 72.84 | 23.12 |
| CCPS | 72.98 | 8.88 | 7.41 | 63.70 | 18.32 | CCPS | 84.89 | 7.11 | 5.25 | 64.07 | 6.45 | CCPS | 70.53 | 9.02 | 7.26 | 61.07 | 19.43 |
| CORAL | 74.38 | 8.78 | 4.43 | 60.59 | 18.31 | CORAL | 85.01 | 10.27 | 5.02 | 53.96 | 9.75 | CORAL | 72.17 | 8.54 | 4.62 | 65.23 | 19.43 |

*Table 2.* Transfer performance of Probe 1 on four downstream benchmarks: HellaSwag, OpenBookQA, ARC-Challenge, and Math-MC. All three models show accuracy gains and calibration improvements across multiple independent benchmark tests. Blue rows are CORAL-steered results. **Bold** numbers indicate best performance and underlined numbers indicate second best. Relevant other methods shown as comparison are SteerConf (Zhou et al., 2025), Few-Shot Prompting (Eval Harness) (Sutawika et al., 2023), ITI (Li et al., 2023), and CCPS (trained on CommonsenseQA) (Zadrozny & Elkan, 2002; Guo et al., 2017; Khanmohammadi et al., 2025).

### Mistral-7B-Instruct-v0.3 (Transfer Experiments)

| ARC-Challenge | | | | | | HellaSwag | | | | | | OpenBookQA | | | | | | Math-MC | | | | |
|---|---|---|---|---|---|---|---|---|---|---|---|---|---|---|---|---|---|---|---|---|---|---|---|
| Method | Acc ↑ | ECE ↓ | cwECE ↓ | NLL ↓ | Brier ↓ | Method | Acc ↑ | ECE ↓ | cwECE ↓ | NLL ↓ | Brier ↓ | Method | Acc ↑ | ECE ↓ | cwECE ↓ | NLL ↓ | Brier ↓ | Method | Acc ↑ | ECE ↓ | cwECE ↓ | NLL ↓ | Brier ↓ |
| SteerConf | 41.21 | 34.09 | 11.09 | 79.37 | 28.41 | SteerConf | 54.77 | 71.61 | 21.48 | 75.38 | 30.73 | SteerConf | 34.41 | 17.41 | 11.25 | 69.03 | 30.03 | SteerConf | 30.27 | 43.90 | 11.72 | 101.67 | 37.09 |
| Few-Shot | 61.86 | 8.58 | 7.73 | 62.52 | 21.83 | Few-Shot | 81.23 | 32.88 | 15.89 | 67.88 | 24.34 | Few-Shot | 50.40 | 11.40 | 8.80 | 70.18 | 24.50 | Few-Shot | 44.89 | 20.16 | 8.67 | 91.55 | 29.38 |
| ITI | 61.86 | 26.01 | 6.34 | 72.34 | 22.34 | ITI | 74.43 | 12.18 | 8.62 | 63.43 | 21.81 | ITI | 50.23 | 11.40 | 7.91 | 60.50 | 22.48 | ITI | 27.55 | 75.94 | 36.83 | 156.77 | 51.56 |
| CCPS | 63.14 | 24.60 | 7.40 | 85.34 | 28.83 | CCPS | 80.79 | 14.64 | 15.73 | 59.44 | 20.07 | CCPS | 52.00 | 28.92 | 8.10 | 92.09 | 32.97 | CCPS | 44.40 | 34.57 | 2.45 | 105.13 | 37.51 |
| CORAL | 73.46 | 3.65 | 3.68 | 60.13 | 16.28 | CORAL | 82.35 | 3.10 | 2.23 | 45.95 | 14.60 | CORAL | 63.20 | 10.66 | 6.65 | 69.57 | 20.70 | CORAL | 45.65 | 3.20 | 2.78 | 65.42 | 23.12 |

### Qwen2.5-7B-Instruct - Transfer Experiments

| ARC-Challenge | | | | | | HellaSwag | | | | | | OpenBookQA | | | | | | Math-MC | | | | |
|---|---|---|---|---|---|---|---|---|---|---|---|---|---|---|---|---|---|---|---|---|---|---|---|
| Method | Acc ↑ | ECE ↓ | cwECE ↓ | NLL ↓ | Brier ↓ | Method | Acc ↑ | ECE ↓ | cwECE ↓ | NLL ↓ | Brier ↓ | Method | Acc ↑ | ECE ↓ | cwECE ↓ | NLL ↓ | Brier ↓ | Method | Acc ↑ | ECE ↓ | cwECE ↓ | NLL ↓ | Brier ↓ |
| SteerConf | 42.80 | 19.12 | 8.93 | 77.89 | 26.92 | SteerConf | 52.29 | 62.09 | 18.55 | 73.84 | 29.90 | SteerConf | 34.03 | 26.95 | 13.26 | 73.32 | 31.13 | SteerConf | 38.28 | 11.15 | 4.31 | 71.62 | 27.89 |
| Few-Shot | 64.25 | 4.84 | 6.22 | 61.35 | 20.69 | Few-Shot | 77.56 | 28.51 | 13.72 | 66.49 | 23.68 | Few-Shot | 49.80 | 17.64 | 10.37 | 74.54 | 25.40 | Few-Shot | 56.78 | 5.12 | 3.19 | 64.49 | 22.09 |
| ITI | 63.93 | 14.67 | 5.10 | 70.99 | 21.17 | ITI | 71.07 | 10.56 | 11.45 | 61.28 | 22.32 | ITI | 49.80 | 17.64 | 9.32 | 64.26 | 23.31 | ITI | 34.85 | 19.29 | 13.55 | 80.57 | 38.76 |
| CCPS | 62.37 | 8.55 | 6.49 | 65.21 | 22.98 | CCPS | 76.61 | 11.45 | 12.94 | 55.97 | 18.59 | CCPS | 47.60 | 15.13 | 10.46 | 72.60 | 26.53 | CCPS | 55.74 | 22.58 | 3.71 | 80.57 | 29.32 |
| CORAL | 74.40 | 4.09 | 3.21 | 56.34 | 16.40 | CORAL | 81.41 | 10.43 | 9.59 | 49.80 | 16.06 | CORAL | 65.40 | 7.21 | 7.44 | 66.11 | 20.92 | CORAL | 58.04 | 3.22 | 3.06 | 63.31 | 21.89 |

### DeepSeek-7B-Chat - Transfer Experiments

| ARC-Challenge | | | | | | HellaSwag | | | | | | OpenBookQA | | | | | | Math-MC | | | | |
|---|---|---|---|---|---|---|---|---|---|---|---|---|---|---|---|---|---|---|---|---|---|---|---|
| Method | Acc ↑ | ECE ↓ | cwECE ↓ | NLL ↓ | Brier ↓ | Method | Acc ↑ | ECE ↓ | cwECE ↓ | NLL ↓ | Brier ↓ | Method | Acc ↑ | ECE ↓ | cwECE ↓ | NLL ↓ | Brier ↓ | Method | Acc ↑ | ECE ↓ | cwECE ↓ | NLL ↓ | Brier ↓ |
| SteerConf | 46.09 | 24.64 | 14.72 | 86.28 | 30.81 | SteerConf | 51.53 | 64.20 | 19.31 | 76.62 | 31.40 | SteerConf | 32.80 | 26.15 | 14.76 | 83.71 | 30.48 | SteerConf | 26.94 | 28.55 | 8.14 | 81.78 | 32.46 |
| Few-Shot | 54.18 | 6.20 | 10.26 | 67.96 | 23.68 | Few-Shot | 76.43 | 29.48 | 14.28 | 68.99 | 24.87 | Few-Shot | 48.00 | 17.12 | 11.55 | 85.11 | 24.87 | Few-Shot | 39.95 | 13.11 | 6.02 | 73.64 | 25.71 |
| ITI | 52.68 | 18.80 | 8.41 | 78.64 | 24.23 | ITI | 70.03 | 10.92 | 7.75 | 64.47 | 22.28 | ITI | 46.48 | 17.12 | 10.38 | 73.37 | 22.82 | ITI | 24.52 | 49.39 | 25.57 | 126.10 | 45.12 |
| CCPS | 54.35 | 15.13 | 10.41 | 74.82 | 27.08 | CCPS | 74.11 | 59.35 | 13.36 | 148.06 | 54.10 | CCPS | 47.60 | 19.57 | 11.43 | 80.47 | 28.73 | CCPS | 39.93 | 40.93 | 3.58 | 111.61 | 41.08 |
| CORAL | 63.82 | 5.45 | 6.90 | 66.76 | 19.83 | CORAL | 81.98 | 11.37 | 5.74 | 42.15 | 13.29 | CORAL | 62.80 | 11.47 | 8.19 | 81.25 | 23.19 | CORAL | 43.73 | 6.46 | 3.37 | 66.21 | 23.47 |

layers, then marginally declines in the final layers. Calibration follows a similar trajectory. ECE drops sharply from early layers and remains consistently low through layers 19–24 before rising slightly at the final layers.

This pattern aligns with prior work on representation formation in transformers (Sun et al., 2025). Early layers primarily handle syntactic processing and token-level features, while later layers commit to specific output tokens. The middle layers, where CORAL performs best, correspond

to the stage where abstract semantic representations are most developed but output distributions are still somewhat malleable.

### 4.3. CORAL's Transferability

A central claim of this work is that CORAL captures generalizable correctness directions rather than task-specific artifacts. Table 2 evaluates this claim through transfer experiments on the complete published test sets of HellaSwag, OpenBookQA, ARC-Challenge, and Math-MC.

CORAL demonstrates strong transferability across all four benchmark test sets and three model families, consistently exceeding baseline methods. On Mistral-7B-Instruct, CORAL achieves the highest accuracy on all transfer tasks while simultaneously improving calibration. The pattern holds for other models, where CORAL consistently achieves competitive or best performance across metrics.

Analyzing transfer patterns reveals that performance gains are strongest when source and target tasks share reasoning structure. Transfer to ARC-Challenge, which requires reasoning similar to CommonsenseQA, shows robust improvements: CORAL achieves 73.46% accuracy on Mistral compared to 61.86% for ITI and 63.14% for CCPS. Transfer to Math-MC also performs well, suggesting that correctness signals generalize across domains requiring multi-step reasoning.

Transfer to HellaSwag presents a more nuanced picture. While CORAL achieves the best accuracy (81.98% on DeepSeek), the ECE improvements are only second-best. This likely reflects a distributional shift: HellaSwag's answer options are substantially longer than those in probe training, making the probe's learned correctness direction less directly applicable.

Notably, CCPS shows limited transfer performance despite strong in-distribution results, underperforming Few-Shot prompting and CORAL baselines on most transfer tasks. This contrast highlights that CORAL's regularized training objective successfully encourages learning of transferable representations rather than task-specific patterns.

## 5. Sparse Autoencoders Analysis

Having demonstrated CORAL's improvements, we investigate why a regularized probe captures correctness information effectively. We use Sparse Autoencoders (SAEs) to decompose activations into mono-semantic features and test whether isolated features can causally influence accuracy and calibration.

### 5.1. SAE Methodology

We train SAEs to decompose model activations into approximately mono-semantic features. The SAE encoder maps a z-score normalized activation $\mathbf{z} \in \mathbb{R}^d$ to a sparse code $\mathbf{f} \in \mathbb{R}^D$ where $D > d$:

$$\mathbf{f} = \mathrm{ReLU}\left(\mathbf{W}_{\mathrm{enc}}\mathbf{z} + \mathbf{b}_{\mathrm{enc}}\right) \tag{11}$$

The decoder reconstructs the activation from the sparse code:

$$\hat{\mathbf{z}} = \mathbf{W}_{\mathrm{dec}}\mathbf{f} + \mathbf{b}_{\mathrm{dec}} \tag{12}$$

We train the SAE to minimize reconstruction error while encouraging sparsity through an L1 penalty weighted by decoder column norms:

$$\mathcal{L}_{\mathrm{SAE}} = \|\mathbf{z} - \hat{\mathbf{z}}\|_2^2 + \lambda \sum_{j=1}^{D} |f_j| \cdot \|\mathbf{d}_j\|_2 \tag{13}$$

where $\mathbf{d}_j$ denotes the $j$-th column of $\mathbf{W}_{\mathrm{dec}}$. We train SAEs at expansion ratios of $4\times$, $8\times$, and $16\times$ for a base dimension of $d = 4,096$. We provide detailed baseline implementation descriptions in Appendix A.1.

We evaluate selected features through causal ablation experiments. For each feature $j$, we remove its contribution from the reconstructed activation:

$$\mathbf{z}_{\mathrm{ablated}} = \hat{\mathbf{z}} - f_j\mathbf{d}_j \tag{14}$$

We use activation patching via forward hooks to replace the target layer's output with the ablated activation and measure the resulting change in accuracy and calibration. Features where ablation increases ECE are beneficial for calibration, while features where ablation decreases ECE are harmful.

### 5.2. SAE Ablation Results

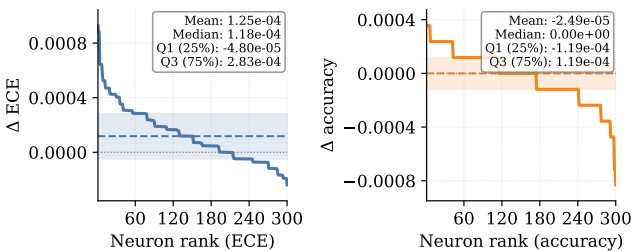

*Figure 3.* Distribution of single-sae-feature ablation impacts on ECE (left) and accuracy (right) for 300 SAE features selected by activation frequency and correlation with residual correctness. Shaded boxes show interquartile ranges and horizontal lines mark medians. Individual features produce negligible causal effects.

Individual SAE features produce negligible causal effects on model calibration. Figure 3 displays the distribution of ablation impacts across 300 SAE features selected by correlation and activation-magnitude filtering. The mean

ECE change from single-feature ablation equals $1.25 \times 10^{-4}$, equivalent to 0.0125 percentage points, negligible compared to CORAL's 15+ percentage point improvements. The interquartile range spans $-4.80 \times 10^{-5}$ to $2.83 \times 10^{-4}$. Although we identify features that individually improve calibration when ablated (harmful features) and features that worsen calibration when ablated (beneficial features), their individual causal impacts remain negligible.

Aggregating beneficial features for steering also fails to recover meaningful gains. Steering with the top 50 to 300 features ranked by ablation impact does not improve accuracy or calibration and often degrades both metrics. For example, steering with the top 128 features yields a 0.12 percentage point decrease in accuracy and 0.14 percentage point increase in ECE. This pattern holds across feature counts and selection criteria.

Existing sparse steering methods show similarly limited effectiveness. Applying CorrSteer (Cho et al., 2025) to DeepSeek-LLM-7B-Chat yields 51.0% accuracy (matching baseline) and 22.01% ECE (versus 22.12% baseline) for MMLU test subset, a statistically insignificant improvement. The contrast with CORAL's substantial gains suggests that individual SAE features capture narrow semantic concepts correlated with correctness but do not encode a coherent correctness direction amenable to steering.

## 6. Discussion

**Regularization Captures What Sparse Methods Miss.** CORAL's performance stems from weight-decay regularization aggregating weak, distributed signals into coherent steering vectors. Our SAE ablations show that individual features produce negligible causal effects on MCQA correctness and aggregating top features by ablation impact fails to recover meaningful gains. This aligns with findings that correctness information exists as a collective property of the activation space (Lindsey et al., 2025), requiring supervised aggregation rather than sparse feature selection.

**Transfer Results Suggest General Correctness Signals.** Strong transfer across four complete independent benchmark test sets suggests that CORAL captures task-general correctness representations rather than dataset-specific patterns. Probes trained on CommonsenseQA and RACE transfer to the complete published test sets of ARC-Challenge, HellaSwag, OpenBookQA, and Math-MC without adaptation, spanning commonsense, science, abstraction, and mathematical reasoning. CCPS shows limited transfer despite strong in-distribution results, possibly because its contrastive training captures dataset-specific distinctions. CORAL's residual prediction objective may encourage learning features that generalize across reasoning types.

**Practical Efficiency for MCQA Deployment.** CORAL trains in under 5 hours on a single NVIDIA RTX 2080 Ti with 8.4k–10k labeled questions, operating only on cached activations with no backpropagation through the base model. At inference time, CORAL adds 0.3% latency overhead relative to the base LLM forward pass: activation extraction, probe inference, and probability correction together add roughly 20 ms per question on an RTX 2080 Ti, negligible compared to the forward pass itself. This overhead is comparable to ITI, which also require activation extraction. The concentration of correctness signals in middle layers (17–21) in our experiments suggests practitioners may reduce computation further by targeting a single layer. A complete timing breakdown is provided in Appendix B.6.

## 7. Conclusion

We present three key findings in this paper where we introduce CORAL: Correctness-Optimized Residual Activation Lens. First, regularized probes recover correctness signals from mid-layer activations reliably. MLP probes achieve accuracy gains of up to 20 points and ECE reductions of up to 14 points, with improvements generalizing across all three tested models. Second, learned steering directions transfer across tasks without retraining. On complete published benchmark test sets, the CORAL probe trained on CommonsenseQA and RACE activations averages 14% accuracy improvements and 49% ECE improvements. These improvements suggest CORAL identifies a general-purpose correctness subspace rather than task-specific decision boundaries. Third, correctness signals require supervised aggregation. Individual SAE features produce negligible causal effects and SAE-based steering degrades performance, while ridge and MLP probes successfully combine distributed signals into coherent steering vectors. CORAL requires only 5 hours of probe training on a single GPU and applies at inference time without modifying model weights.

## 8. Limitations

CORAL's evaluation is limited to multiple-choice question answering, leaving open whether correctness signals generalize to free-form generation where correctness is harder to define. Within MCQA, our transfer evaluation is out-of-sample (zero question overlap with training data) but not fully out-of-domain: all benchmarks involve reasoning-based multiple-choice tasks, and we do not test on radically different formats such as open-ended generation, multi-hop retrieval, or non-English benchmarks. Our results demonstrate that CORAL's correctness subspace transfers across reasoning types within the MCQA format, but we do not claim arbitrary domain transfer.

## Impact Statement

CORAL contributes to making large language model outputs more reliable. Improved calibration could benefit high-stakes domains such as medical diagnosis, legal analysis, and scientific research, where calibrated confidence enables systems to flag uncertain outputs for human review.

However, the same methodology could be adapted for harmful purposes. Malicious actors could train probes to steer models toward misinformation or suppress particular viewpoints, and the transferability we demonstrate means harmful steering vectors could generalize across tasks. We release our code and probe weights to enable reproducibility and encourage research into detection mechanisms for adversarial steering.

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

# A. Appendix

### A.1. Sparse Autoencoder Implementation Details

**Feature Selection via Correlation Filtering.** Inspired by CorrSteer (Cho et al., 2025), we identify calibration-relevant features by computing Pearson correlation between each SAE feature's activation and the residual correctness label. We first filter features by activation frequency, requiring a minimum of 10 active samples and activation frequency above 0.01%. We define a feature as active when its magnitude exceeds $10^{-4}$. From the filtered set, we select the top-$k$ features with the highest positive correlation with the target variable. This correlation-based selection identifies features that linearly associate with calibration behavior.

**Feature Selection via Activation Magnitude.** We implement an alternative selection method based on the feature's expected impact on downstream predictions. For each feature $j$, we compute its directional sensitivity as the alignment between its decoder column and the ridge probe weight vector:

$$s_j = (\mathbf{d}_j \odot \boldsymbol{\sigma})^\top \mathbf{w} \tag{15}$$

where $\odot$ denotes elementwise multiplication and $\boldsymbol{\sigma}$ is the activation standard deviation. We multiply this sensitivity by the mean feature activation to obtain an approximate impact score. We select features with the largest absolute impact scores, capturing directions that most influence the probe's residual prediction.

**SAE Feature Steering.** We implement SAE-based steering using forward hooks to modify hidden states during inference. We first identify beneficial features from the SAE ablation analysis results, where a feature is considered beneficial if ablating it decreases accuracy (negative accuracy impact) or increases ECE (positive ECE impact). For each beneficial feature $j$, we compute a steering weight:

$$w_j = \alpha_{\text{acc}} \cdot \max(-\Delta_{\text{acc},j}, 0) + \alpha_{\text{cal}} \cdot \max(\Delta_{\text{ECE},j}, 0) \tag{16}$$

where $\Delta_{\text{acc},j} = \text{Acc}_{\text{ablated},j} - \text{Acc}_{\text{baseline}}$ is the change in accuracy when ablating feature $j$, and $\Delta_{\text{ECE},j} = \text{ECE}_{\text{ablated},j} - \text{ECE}_{\text{baseline}}$ is the corresponding change in calibration error. The hyperparameters $\alpha_{\text{acc}}$ and $\alpha_{\text{cal}}$ control the relative importance of accuracy and calibration contributions. We normalize these weights to sum to one across all selected features.

During inference, we register a forward hook at the target layer that applies additive steering. For each hidden state $\mathbf{h}$, we normalize using SAE training statistics, encode to obtain feature activations $f_j$, and compute the steering perturbation:

$$\mathbf{h}' = \mathbf{h} + \gamma \cdot \left( \sum_{j \in \mathcal{S}} f_j \cdot w_j \cdot \mathbf{d}_j \right) \odot \boldsymbol{\sigma} \tag{17}$$

where $\mathcal{S}$ is the set of beneficial features, $\mathbf{d}_j$ is the decoder column for feature $j$, $\boldsymbol{\sigma}$ is the activation standard deviation, and $\gamma$ controls the global steering strength. This intervention amplifies features that causally improve calibration while preserving accuracy.

### A.2. Attention-Head Activations Analysis

To investigate whether calibration-relevant information is localized to specific attention heads—as would be required for Inference-Time Intervention (ITI) style surgical steering —we conducted a systematic probe analysis across all attention heads in the model.

**Methodology.** We extracted activations from the output of each attention head (before the output projection) at the last token position of each answer option. For DeepSeek-7B-Chat, this yields 960 total head representations (30 layers × 32 heads), each with dimension $d_{\text{head}} = 128$. Following the ITI approach, we trained separate 4-layer MLP probes on each (layer, head) pair to predict the residual correctness signal $r_i = y_i - \hat{p}_i$. Each probe uses hidden dimensions $(256, 128, 64, 32)$ with ReLU activations and dropout regularization. We evaluated generalization performance using 5-fold GroupKFold cross-validation (grouped by question to prevent leakage) and report $R^2$ scores on a held-out validation set.

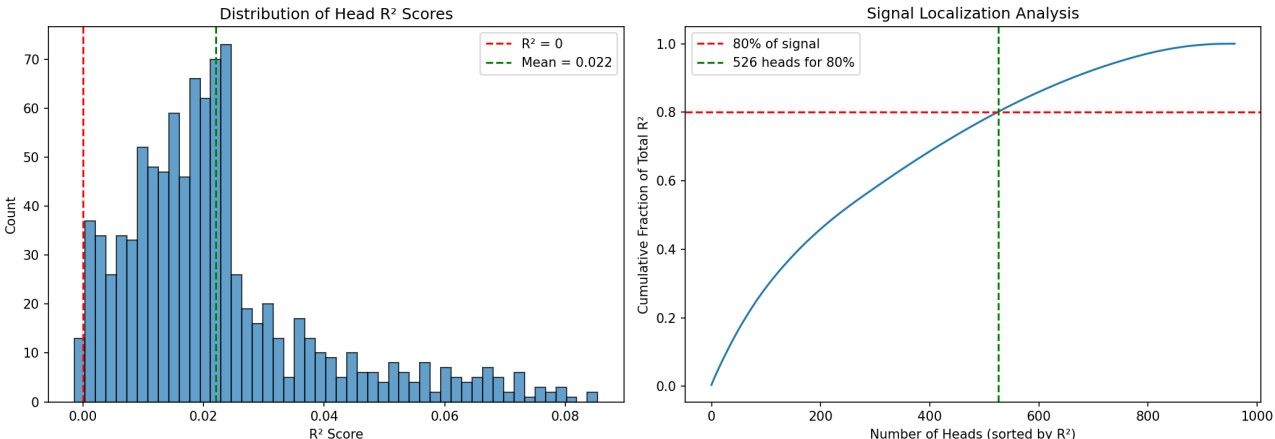

*Figure 4.* Distribution of calibration signal across attention heads. *Left:* Histogram of $R^2$ scores from 4-layer MLP probes trained on individual attention head activations to predict residual correctness. The mean $R^2 = 0.022$ and maximum $R^2 = 0.085$, with no head exceeding $R^2 > 0.10$. *Right:* Cumulative signal analysis showing that 526 heads (55% of all 960 heads) are required to capture 80% of the total predictive signal, indicating that calibration information is distributed across the attention mechanism rather than localized to specific heads.

**Results.** Figure 4 shows the distribution of $R^2$ scores across all 960 attention heads. The left panel displays a histogram revealing that most heads have low predictive power, with a mean $R^2 = 0.022$ and maximum $R^2 = 0.085$. Notably, no individual head achieves $R^2 > 0.10$, and only 82 heads (8.5%) exceed $R^2 > 0.05$.

The right panel presents a signal localization analysis: when heads are sorted by $R^2$ and their contributions are accumulated, we find that 526 heads (55% of all heads) are required to capture 80% of the total predictive signal. Equivalently, the top-48 heads—a reasonable target for surgical intervention—contain only 15.8% of the total positive $R^2$. This indicates a highly distributed representation where calibration information is spread diffusely across the attention mechanism rather than concentrated in a small subset of "calibration heads."

### A.3. Signal Dimensionality Analysis

To characterize the geometric structure of the calibration signal in activation space, we analyze how predictive power grows with the dimensionality of the representation. Specifically, we apply PCA to reduce the $d_{\text{model}} = 4096$ dimensional residual stream activations to $k$ principal components, then train Ridge regression probes to predict the residual correctness signal $r_i = y_i - \hat{p}_i$.

**Methodology.** For each layer, we fit PCA on the training activations and evaluate cross-validated $R^2$ as a function of the number of retained components $k \in \{1, 2, \ldots, 100\}$. If the calibration signal were low-dimensional (e.g., rank-1 or rank-10), we would expect $R^2$ to saturate quickly with few components. Conversely, a gradual increase in $R^2$ with $k$ indicates that the signal is distributed across many directions in activation space.

**Results.** Figure 5 shows the results averaged across groups of 5 consecutive layers. The left panel reveals that $R^2$ grows gradually with the number of PCA components across all layer groups, without saturating even at 100 components. Later layers (L20–24, L25–29) achieve higher maximum $R^2$ values (0.06–0.07), consistent with calibration information accumulating in deeper layers. The right panel shows cumulative explained variance: early layers (L0–4) have highly concentrated variance where 3 components capture over 90% of activation variance, while later layers exhibit more distributed representations.

Importantly, despite the high concentration of *activation variance* in few directions for early layers, the *calibration signal* remains distributed—high explained variance does not imply high predictive power in those directions. This suggests that calibration-relevant information is encoded in subtle, distributed patterns rather than dominant principal components, motivating the use of full-dimensional MLP probes rather than low-rank approximations.

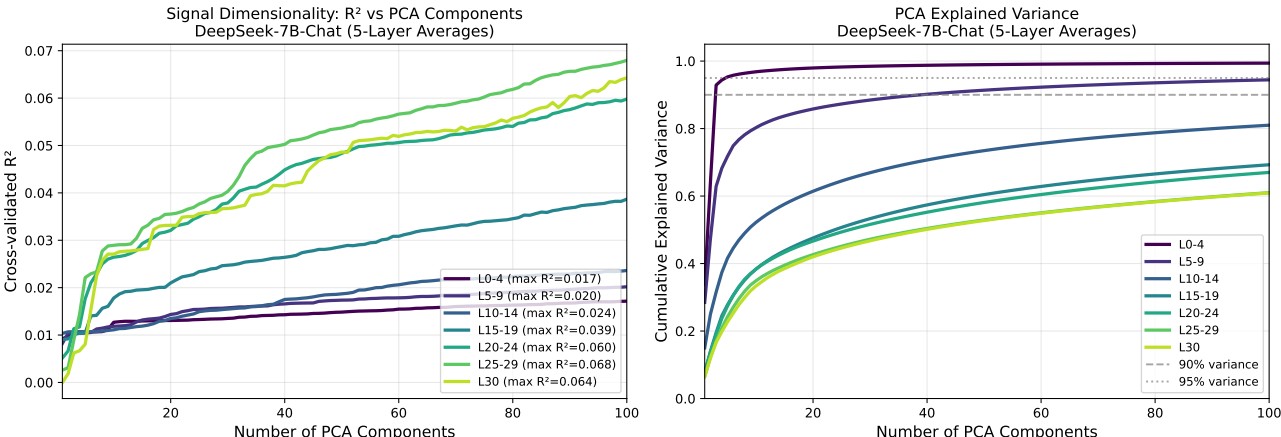

*Figure 5.* **Signal dimensionality analysis across layers.** *Left:* Cross-validated $R^2$ for predicting residual correctness as a function of the number of PCA components, averaged over 5-layer groups. Later layers (L20–29) achieve higher predictive power, but all layer groups show gradual $R^2$ growth without saturation, indicating that the calibration signal is distributed across many dimensions rather than concentrated in a low-rank subspace. *Right:* Cumulative explained variance of PCA components. Early layers exhibit highly concentrated activation variance (3 components capture >90%), while later layers have more distributed representations. The mismatch between variance concentration and predictive power suggests calibration information resides in subtle activation patterns rather than dominant principal directions.

## B. Baseline Implementation Details

### B.1. Inference-Time Intervention (ITI) for Attention Head Steering

ITI (Li et al., 2023) is an activation steering method that identifies and modifies specific attention head outputs during inference to improve model truthfulness.

**Stage 1: Activation Collection.** For each prompt $x$ in a binary-labeled dataset (e.g., TruthfulQA with correct/incorrect answer labels), we extract attention head activations at the last token position. Let $\mathbf{h}_{\ell,k} \in \mathbb{R}^{d_{\text{head}}}$ denote the output of attention head $k$ at layer $\ell$. We collect activations across all layers and heads using the output projection input:

$$\mathbf{H} = \{\mathbf{h}_{\ell,k}^{(i)}\}_{\ell=1,k=1,i=1}^{L,K,N} \tag{18}$$

where $L$ is the number of layers, $K$ is the number of heads per layer, and $N$ is the number of samples.

**Stage 2: Probe Training and Head Selection.** For each attention head $(\ell, k)$, we train a logistic regression probe to classify truthful vs. untruthful responses:

$$p(y = 1 \mid \mathbf{h}_{\ell,k}) = \sigma(\mathbf{w}_{\ell,k}^{\top} \mathbf{h}_{\ell,k} + b_{\ell,k}) \tag{19}$$

where $\mathbf{w}_{\ell,k} \in \mathbb{R}^{d_{\text{head}}}$ is the learned weight vector. The top-$M$ heads are selected based on validation accuracy, yielding $\mathcal{T} = \{(\ell_1, k_1), \ldots, (\ell_M, k_M)\}$.

**Stage 3: Intervention Direction Computation.** For each selected head, we compute the *mass mean shift* direction as the difference between class centroids:

$$\boldsymbol{\theta}_{\ell,k} = \frac{1}{|\mathcal{D}_+|} \sum_{i \in \mathcal{D}_+} \mathbf{h}_{\ell,k}^{(i)} - \frac{1}{|\mathcal{D}_-|} \sum_{i \in \mathcal{D}_-} \mathbf{h}_{\ell,k}^{(i)} \tag{20}$$

where $\mathcal{D}_+$ and $\mathcal{D}_-$ denote the sets of truthful and untruthful samples, respectively. The direction is normalized: $\hat{\boldsymbol{\theta}}_{\ell,k} = \boldsymbol{\theta}_{\ell,k}/\|\boldsymbol{\theta}_{\ell,k}\|_2$.

**Stage 4: Inference-Time Steering.** During inference, we modify the attention head outputs at the last token position for all selected heads:

$$\tilde{\mathbf{h}}_{\ell,k} = \mathbf{h}_{\ell,k} + \alpha \cdot \hat{\boldsymbol{\theta}}_{\ell,k} \tag{21}$$

where $\alpha$ is a scalar hyperparameter controlling intervention strength. The intervention is applied using PyVene (Wu et al., 2024) by hooking into `model.layers[ℓ].self_attn.o_proj.input`.

### B.2. Confidence-Conditioned Prediction with Shifting (CCPS)

CCPS (Khanmohammadi et al., 2025) is a two-stage framework that learns to estimate response correctness by analyzing perturbation-based features extracted from model internals.

**Stage 1: Perturbation-Based Feature Extraction.** For each generated token $t$ at position $i$, we extract the hidden state $\mathbf{h}_i \in \mathbb{R}^d$ and logits $\mathbf{l}_i \in \mathbb{R}^V$ from the final layer before the language model head. We compute the Jacobian vector:

$$\mathbf{J}_t = \nabla_{\mathbf{h}_i} \log p(t \mid \mathbf{h}_i) \tag{22}$$

We generate a sequence of perturbed hidden states along the Jacobian direction:

$$\tilde{\mathbf{h}}_i^{(\epsilon)} = \mathbf{h}_i + \epsilon \cdot \frac{\mathbf{J}_t}{\|\mathbf{J}_t\|_2}, \quad \epsilon \in \{\epsilon_1, \ldots, \epsilon_K\} \tag{23}$$

and compute corresponding perturbed logits $\tilde{\mathbf{l}}_i^{(\epsilon)} = \text{LMHead}(\tilde{\mathbf{h}}_i^{(\epsilon)})$.

From these, we extract comprehensive features including:

- **Original state features**: log-probability, entropy, top-1/top-2 margin, hidden state norm
- **Perturbation sensitivity**: $\epsilon$-to-flip (minimum perturbation to change argmax), Jacobian norm
- **PEI (Perturbation Expected Information)**:

$$\text{PEI} = \frac{1}{K} \sum_{k=1}^{K} \max \left( 0, \log p(t \mid \mathbf{h}_i) - \log p(t \mid \tilde{\mathbf{h}}_i^{(\epsilon_k)}) \right) \tag{24}$$

- **Divergence metrics**: KL divergence, JS divergence, cosine similarity between original and perturbed distributions

**Stage 2: Contrastive Embedding Learning.** Let $\mathbf{f} \in \mathbb{R}^{d_f}$ denote the concatenated feature vector. We train an embedding network $g_\phi : \mathbb{R}^{d_f} \to \mathbb{R}^{d_e}$ using max-margin contrastive loss:

$$\mathcal{L}_{\text{cont}} = \frac{1}{|\mathcal{B}|} \sum_{(i,j) \in \mathcal{B}} (1 - y_{ij}) \cdot d_{ij}^2 + y_{ij} \cdot \max(0, m - d_{ij})^2 \tag{25}$$

where $d_{ij} = \|g_\phi(\mathbf{f}_i) - g_\phi(\mathbf{f}_j)\|_2$, $y_{ij} = \mathbb{1}[y_i \neq y_j]$ indicates whether samples have different correctness labels, and $m$ is the margin hyperparameter.

The embedding network architecture is:

$$g_\phi(\mathbf{f}) = \text{Linear}_{d_e}(\text{ReLU}(\text{Linear}_{64}(\text{ReLU}(\text{Linear}_{64}(\mathbf{f}))))) \tag{26}$$

with dropout ($p = 0.1$) after each activation.

**Stage 3: Correctness Classification.** A classifier head is trained on top of the frozen or fine-tuned embeddings:

$$p(y = 1 \mid \mathbf{f}) = \text{softmax}(h_\psi(g_\phi(\mathbf{f}))) \tag{27}$$

where $h_\psi$ is an MLP with hidden dimensions [48, 24, 12]. Training uses cross-entropy loss with AdamW optimizer ($\text{lr} = 10^{-4}$, weight decay $= 0.01$).

**Inference.** For a new response, features are extracted from all generated tokens, standardized using the training scaler, and passed through the trained model. The probability $p(y = 1 \mid \mathbf{f})$ serves as the confidence estimate.

### B.3. Confidence Steering via Prompt Shifting (SteerConf)

SteerConf (Zhou et al., 2025) modulates LLM verbalized confidence through carefully designed prompt instructions that shift the model's risk tolerance.

**Prompt Design.** The base prompt template requests both an answer and a numerical confidence score:

```
Read the question, analyze step by step, provide your answer and your confidence
in this answer.  Note:  The confidence indicates how likely you think your answer
is true.
Use the following format to answer:
```Explanation:  [insert step-by-step analysis here]
Answer and Confidence (0-100):  [ONLY the {answer_type}], [confidence]%```
```

**Confidence Shifting Instructions.** We inject steering instructions that modulate the model's confidence calibration:

- **Very Cautious**: "You are making important decisions, thus you should avoid giving a wrong answer with high confidence. You should be very cautious, and tend to give low confidence on almost all of the answers."
- **Cautious**: "You are making important decisions, thus you should avoid giving a wrong answer with high confidence."
- **Confident**: "You are making important decisions, thus you should avoid giving a right answer with low confidence."
- **Very Confident**: "You are making important decisions, thus you should avoid giving a right answer with low confidence. You should be very confident, and tend to give high confidence on almost all of the answers."

**Multi-Query Aggregation.** For each question, we query the model $N$ times with temperature $\tau > 0$ to obtain answer-confidence pairs $\{(a_i, c_i)\}_{i=1}^{N}$. We compute aggregated metrics:

**Consistency Score**: The frequency of the most common answer:

$$S_{\text{cons}} = \frac{|\{i : a_i = \hat{a}\}|}{N}, \quad \text{where } \hat{a} = \arg\max_a |\{i : a_i = a\}| \tag{28}$$

**Weighted Confidence**: Combining mean confidence with consistency:

$$S_{\text{weighted}} = \bar{c} \cdot (1 - V_{\text{ans}}) \cdot (1 - V_{\text{conf}}) \tag{29}$$

where $\bar{c} = \frac{1}{N} \sum_i c_i$ is the mean confidence, $V_{\text{ans}} = 1 - S_{\text{cons}}$ is answer variability, and $V_{\text{conf}} = \frac{\sigma_c / \bar{c}}{1 + \sigma_c / \bar{c}}$ is the normalized coefficient of variation of confidence scores.

**Final Answer Selection.** With majority voting, the final answer is selected as $\hat{a}$. If there are ties, we select the answer with the highest mean confidence among tied candidates.

### B.4. Hyperparameters and Training Details

**ITI.** We select the top $M = 48$ attention heads based on probe validation accuracy. The intervention strength $\alpha$ is tuned on a held-out validation set, typically in the range $[1.0, 3.0]$. Probes are trained with scikit-learn's `LogisticRegression` with `max_iter=1000`.

**CCPS.** Contrastive model: hidden dimensions [64, 64], embedding dimension 8, ELU activation, dropout 0.05, margin $m = 1.0$, trained for 5000 steps with batch size 64, learning rate $10^{-4}$. Classifier: hidden dimensions [48, 24, 12], trained for 5000 steps with learning rate $10^{-4}$. Feature extraction uses perturbation radii $\epsilon \in \{0.01, 0.1, 0.5, 1.0, 2.0, 5.0, 10.0, 20.0\}$.

**SteerConf.** Ensemble size $N = 5$ with temperature $\tau = 0.7$. We use chain-of-thought prompting for reasoning tasks. Confidence scores are normalized to $[0, 1]$ by dividing by 100.

## B.5. Baseline Tuning and Validation Protocols

To ensure a fair comparison, all baseline methods were granted equivalent tuning flexibility to CORAL's per-model and per-dataset layer/$\gamma$ selection. We describe the tuning protocols for each method below.

**ITI (Inference-Time Intervention).**     For each model and target dataset combination, we performed the following tuning:

- **Head selection**: Logistic regression probes were trained on a held-out validation split (20% of training data) to select the top-$M$ attention heads. We tuned $M \in \{24, 48, 96\}$ per model.
- **Intervention strength**: The multiplier $\alpha$ was selected via grid search over $\{0.5, 1.0, 1.5, 2.0, 2.5, 3.0\}$ using validation ECE as the selection criterion.
- **Direction computation**: Both probe-weight directions and mass-mean-shift directions were evaluated; the better-performing variant was selected per model.

This mirrors the per-model tuning afforded to CORAL for layer selection and steering magnitude $\gamma$.

**CCPS (Confidence-Conditioned Prediction with Shifting).**     CCPS was tuned with the following protocol:

- **Training data**: The contrastive embedding model and classifier were trained on a source dataset (e.g., CT-CHOICE) with an 80/20 train/validation split.
- **Architecture search**: Hidden dimensions for both the embedding network and classifier head were tuned per model using validation AUROC. We searched over embedding dimensions $\{8, 16, 32\}$ and classifier hidden layers from $\{[48, 24], [48, 24, 12], [64, 32, 16]\}$.
- **Per-dataset adaptation**: When transferring to a new target dataset, we allowed optional fine-tuning of the classifier head on a small validation subset (up to 500 samples) while keeping the contrastive encoder frozen.
- **Feature selection**: The perturbation radii $\{\epsilon_k\}$ and included feature types were validated per model to ensure stable feature extraction across different model architectures.

**SteerConf (Confidence Steering via Prompt Shifting).**     SteerConf hyperparameters were tuned per model and dataset:

- **Prompt variant selection**: The steering prompt type (vanilla, cautious, very_cautious, confident, very_confident) was selected based on validation ECE for each model-dataset pair.
- **Sampling parameters**: Temperature $\tau \in \{0.5, 0.7, 1.0\}$ and ensemble size $N \in \{1, 3, 5, 10\}$ were tuned on a validation split.
- **Aggregation method**: The confidence aggregation strategy (mean, weighted, consistency-adjusted) was selected per dataset based on validation calibration metrics.
- **Chain-of-thought**: Whether to use CoT prompting was determined per dataset based on task complexity and validation performance.

**Summary.**     All baseline methods were allowed comparable tuning flexibility to CORAL. Specifically, each method could select its key hyperparameters (ITI: heads and $\alpha$; CCPS: architecture and features; SteerConf: prompt type and sampling) independently for each model and dataset combination using held-out validation data. This ensures that performance differences reflect fundamental methodological advantages rather than tuning disparities.

## B.6. Inference Overhead Analysis

We measure CORAL's inference overhead on DeepSeek-7B-Chat (MMLU, 97 questions, 3 warmup rounds) using a single NVIDIA RTX 2080 Ti (10.6 GB VRAM). Table 3 reports per-question timing statistics.

CORAL adds 20.8 ms (0.3%) per question relative to the vanilla forward pass. This overhead decomposes into activation extraction (3.95 ms per option via forward hooks, requiring no additional forward pass), probe inference (1.89 ms median per question), and probability correction (0.05 ms per question). The bottleneck is the LLM forward pass at approximately 1.76 s per option. ITI has comparable activation-hook overhead, whereas SteerConf incurs overhead through additional prompted generations. CORAL's additional cost is the MLP forward pass, which replaces ITI's per-head direction lookup and SteerConf's confidence-based heuristic.

Training cost is also modest: probe training operates on cached activations with no backpropagation through the base model, completing in under 5 hours on a single RTX 2080 Ti.

*Table 3.* Inference timing breakdown for CORAL on DeepSeek-7B-Chat (MMLU). All times in milliseconds.

| Stage | Mean | Std | Median |
|---|---|---|---|
| Vanilla forward (per option) | 1759.23 | 41.40 | 1745.27 |
| Extract forward (per option) | 1763.18 | 43.16 | 1747.54 |
| Probe inference (per question) | 4.18 | 22.42 | 1.89 |
| Correction (per question) | 0.05 | 0.01 | 0.05 |
| Total vanilla (per question) | 7041.01 | 162.30 | 6975.22 |
| Total CORAL (per question) | 7061.81 | 169.70 | 6991.00 |

## B.7. Linear Probe Ablation

To isolate the contribution of nonlinear capacity in the probe architecture, we compare CORAL's MLP probe against ridge regression (a linear probe with L2 regularization) across all three models on MMLU. Table 4 reports best-layer results for each model.

*Table 4.* Comparison of ridge regression and MLP probes on MMLU (best layer per model). Ridge achieves competitive accuracy but substantially worse calibration, confirming that nonlinear capacity is important for calibration correction.

| Model | Probe | Acc ↑ | ECE ↓ | Brier ↓ | cwECE ↓ |
|---|---|---|---|---|---|
| DeepSeek-7B | Ridge | 0.530 | 0.170 | 0.245 | 0.094 |
| | MLP | 0.540 | 0.062 | 0.213 | 0.044 |
| Mistral-7B | Ridge | 0.624 | 0.173 | 0.224 | 0.094 |
| | MLP | 0.642 | 0.033 | 0.179 | 0.028 |
| Qwen-7B | Ridge | 0.734 | 0.171 | 0.201 | 0.091 |
| | MLP | 0.742 | 0.042 | 0.155 | 0.028 |

Three findings emerge. First, ridge accuracy is competitive with the MLP: the accuracy gap is at most 1.8 percentage points, confirming that a meaningful correctness signal is linearly accessible in the residual stream. Second, the MLP achieves $2.5\times$ to $5.2\times$ lower ECE than ridge across all three models. For Mistral-7B, ECE drops from 0.173 (ridge) to 0.033 (MLP). Since calibration improvement is a central contribution of CORAL, this gap justifies the MLP as the primary probe. Third, both probes improve over the unsteered baseline (e.g., DeepSeek baseline ECE is 0.221; ridge achieves 0.170), confirming that correctness information exists at the linear level. However, the MLP's nonlinear capacity extracts substantially more of it, particularly for calibration.

These results tell a clean story: the linear component of the correctness signal helps with accuracy, but calibration correction requires the nonlinear capacity of the MLP.

## B.8. Loss Function Ablation

To isolate the contribution of the residual probability target, we trained the same 4-layer MLP architecture (dimensions 1024, 512, 256, 128) on Layer 19 of DeepSeek-7B-Chat, varying only the training target. Table 5 reports steered performance on MMLU.

*Table 5.* Loss function ablation on DeepSeek-7B-Chat (MMLU, Layer 19). All three targets improve accuracy, but only the residual probability target improves calibration.

| Loss Target | Acc ↑ | △ Acc | ECE ↓ | △ ECE | Brier ↓ | △ Brier |
|---|---|---|---|---|---|---|
| Baseline (no steering) | 0.508 | — | 0.221 | — | 0.263 | — |
| Residual prob. (ours) | 0.530 | +0.022 | 0.062 | −0.159 | 0.207 | −0.056 |
| Binary CE | 0.532 | +0.024 | 0.238 | +0.017 | 0.274 | +0.011 |
| Brier score | 0.525 | +0.017 | 0.229 | +0.008 | 0.279 | +0.016 |

All three losses improve accuracy (+1.7 to +2.4 percentage points), confirming that the probe architecture itself provides some benefit regardless of the target. However, only the residual probability target improves calibration: ECE drops 72% ($0.221 \rightarrow 0.062$) and Brier drops 21%. The binary correctness and Brier score targets worsen calibration despite improving

accuracy.

The residual probability target $r = y - p$ is uniquely effective because it encodes both the direction and magnitude of the needed correction per option. Binary correctness tells the probe "right or wrong" but not "by how much." The Brier score target is always non-negative, losing sign information about whether the model is overconfident or underconfident. Only $r = y - p$ encodes the exact correction vector in probability space, enabling steering to fix miscalibration rather than amplify confidence.

### B.9. Steering Strength $\gamma$ Ablation

Table 6 reports a sweep over the steering strength hyperparameter $\gamma$ on DeepSeek-7B-Chat (MMLU, MLP probe, Layer 21).

*Table 6.* Effect of steering strength $\gamma$ on DeepSeek-7B-Chat (MMLU). $\gamma = 1.0$ achieves the best accuracy while maintaining strong calibration. Higher values overshoot.

| $\gamma$ | Acc $\uparrow$ | ECE $\downarrow$ | Brier $\downarrow$ |
|---|---|---|---|
| 0 (baseline) | 0.508 | 0.221 | 0.263 |
| 0.5 | 0.528 | 0.117 | 0.221 |
| 1.0 | 0.540 | 0.062 | 0.213 |
| 2.0 | 0.495 | 0.072 | 0.218 |
| 5.0 | 0.355 | 0.289 | 0.304 |

$\gamma = 1.0$ is consistently optimal across all three model families in our experiments. This is not coincidental: the residual target $r = y - p$ is bounded in $[-1, 1]$, so $\gamma = 1.0$ corresponds to a unit-magnitude correction in probability space. This natural scaling generalizes to any model whose residuals share this range, which is guaranteed by the target definition. At $\gamma = 2.0$, calibration remains improved but accuracy degrades, and at $\gamma = 5.0$, all metrics deteriorate substantially. We recommend $\gamma = 1.0$ as a strong default, though validation-based tuning is advisable for new architectures or model scales.

