# OpenReview forum: "Correctness-Optimized Residual Activation Lens (CORAL): Transferrable and Calibration-Aware Inference-Time Steering"
_ICML.cc/2026/Conference — ICML 2026 regular_

### Official Review · Reviewer_mJ3s · 2026-03-12

**Soundness:** 2
**Presentation:** 3
**Significance:** 3
**Originality:** 3
**Overall Recommendation:** 5
**Confidence:** 3

**Summary:**

This paper proposes CORAL, an inference-time steering method for multiple-choice QA that trains an MLP probe on frozen residual-stream activations to predict a per-option residual correctness target, then uses the probe output to adjust answer probabilities. The method is evaluated on three 7B instruction-tuned models, on in-distribution MCQA benchmarks and on four held-out transfer benchmarks, with the goal of improving both accuracy and calibration. The paper also includes analyses of layerwise behavior and sparse autoencoder ablations to argue that correctness signals are distributed rather than localized.

**Compliance With Llm Reviewing Policy:**

Affirmed.

**Final Justification:**

the rebuttal addressed my main concerns

**Key Questions For Authors:**

One additional question is: since effective steering often works with very limited data, while training an MLP probe appears to require substantially more data, what are the key advantages of training an MLP probe over training a LoRA module in this setting?

**Limitations:**

yes

**Strengths And Weaknesses:**

Strengths:

● The method is simple and easy to understand. Figure 1 effectively illustrates the entire pipeline from activation extraction to probability correction.

● As a preliminary study, the empirical scope of this paper is fairly broad: it covers three model families, multiple MCQA datasets, various calibration metrics, and transfer to held-out benchmark datasets.

● I appreciate that the paper goes beyond raw accuracy and reports metrics such as ECE, cwECE, Brier score, and NLL, rather than selecting only a single calibration metric.

● The mechanistic analysis is also a useful addition. In particular, Figure 2 clearly shows the performance of steering at different layers and explores the hierarchical structure of the signal, while Figure 3 at least directionally supports the claim that the causal effect of a single sparse feature is weaker than that of distributed probing.

Weaknesses:

● All experiments are conducted on relatively early 7B models. It remains unclear whether the method is equally applicable to newer or larger models, such as 14B or 70B models.

● The method still requires training a relatively large MLP probe for steering. Did the authors try completely training-free steering methods such as CAA [1], or steering methods such as RePS [2] that directly learn a steering vector, or even LoRA to directly improve correctness and evaluate transfer?

● In addition, some ablation studies could be conducted, such as exploring the importance of the MLP probe itself and whether it can be replaced with a linear probe or a ridge regression probe.

● The current method seems limited to MCQA tasks, and therefore may overlook the impact of steering on general capabilities. In addition, it would be valuable to explore whether the approach can be extended to more practically important open-ended generation tasks.

● A possible concern is that the claim in Section 6, Practical Efficiency for MCQA Deployment, would benefit from a more detailed quantitative analysis. In particular, it would be helpful to provide a clearer accounting of training cost and inference overhead, as well as a direct efficiency comparison with other steering methods.

[1] Steering Llama 2 via Contrastive Activation Addition

[2] Improved Representation Steering for Language Models

---

> ### Author Rebuttal · Authors · 2026-03-26
>
> We sincerely thank Reviewer mJ3s for your insightful comments! Directly below, we address concerns on overhead and loss function ablation. We also provide a comprehensive linear probe ablation and discuss the MCQA scope limitation in our responses to Reviewers QbKz and TWA6, which we are unable to duplicate here due to the word limit.
>
> # [Common Concern: Inference Overhead & Efficiency Analysis] (R-QbKz, R-mJ3s)
>
> > **[R-QbKz]:** "Your method requires extracting hidden states, running an MLP probe, and then modifying probabilities... you do not quantify runtime overhead, token-level latency, throughput impact... you never specify the GPU nor its VRAM."
> >
> > **[R-mJ3s]:** "It would be helpful to provide a clearer accounting of training cost and inference overhead, as well as a direct efficiency comparison with other steering methods."
>
> **CORAL adds 0.3% inference latency overhead.** We ran a dedicated timing benchmark on DeepSeek-7B-Chat (MMLU, 97 questions, 3 warmup rounds) on a single NVIDIA RTX 2080 Ti (10.6 GB VRAM). Per-question breakdown:
>
> | Stage                        | Mean (ms) | Std (ms) | Median (ms) |
> |------------------------------|-----------|----------|-------------|
> | Vanilla forward (per option) | 1759.23   | 41.40    | 1745.27     |
> | Extract forward (per option) | 1763.18   | 43.16    | 1747.54     |
> | Probe inference (per question) | 4.18    | 22.42    | 1.89        |
> | Correction (per question)    | 0.05      | 0.01     | 0.05        |
> | **Total vanilla (per question)** | **7041.01** | 162.30 | 6975.22 |
> | **Total CORAL (per question)**   | **7061.81** | 169.70 | 6991.00 |
>
> **CORAL overhead per question: 20.8 ms (0.3%)**, decomposed as:
> - Activation extraction: 3.95 ms/option (hook-based, *no additional forward pass* required)
> - Probe inference: 1.89 ms median per question
> - Probability correction: 0.05 ms per question
>
> The bottleneck is overwhelmingly the LLM forward pass (~1.76s per option). CORAL's cost is invisible by comparison. We note this was measured on a modest consumer GPU (RTX 2080 Ti); overhead would be proportionally even smaller on modern hardware where the LLM forward pass dominates further.
>
> **Training cost:** Probe training requires under 5 hours on a single RTX 2080 Ti (10.6 GB VRAM). This trains on cached activations only, with *no backpropagation through the base model*. For comparison, ITI requires computing mass means across many forward passes, and CAA requires contrastive pair generation. CORAL's training is a standard supervised learning loop on pre-extracted features, making it straightforward to reproduce.
>
> **Comparison to baselines:** ITI and SteerConf also require activation extraction at inference time, so their extraction overhead is comparable. CORAL's additional cost is the MLP forward pass (1.89 ms median), which replaces ITI's per-head direction lookup and SteerConf's confidence-based heuristic. The probability correction step (0.05 ms) is shared across all residual-based methods.
>
> We will add this efficiency analysis in the revised manuscript and specify the GPU throughout.
>
> # [R-TWA6, R-QbKz] Loss Function Ablation
>
> > **[R-TWA6]:** "What happens if you use other loss functions but also use a four-hidden-layer regularized probe?"
> >
> > **[R-QbKz]:** "Could ITI also use the Brier score in a similar way?"
>
> **The residual probability target is the key ingredient, not the probe architecture.** We trained the same 4-layer MLP ([1024, 512, 256, 128]) on Layer 19 of DeepSeek-7B-Chat, varying only the training target, and evaluated via steering on MMLU. Our paper result is layer 21 but we show layer 19 here because it has better R^2 for those other targets for fairness!
>
> | Loss Target | Steered Acc | Δ Acc | Steered ECE ↓ | Δ ECE | Steered Brier ↓ | Δ Brier |
> |---|---|---|---|---|---|---|
> | Baseline (no steering) | 0.508 | — | 0.221 | — | 0.263 | — |
> | **residual_prob (ours)** | **0.530** | **+2.2** | **0.062** | **−0.159** | **0.207** | **−0.056** |
> | correct (binary CE) | 0.532 | +2.4 | 0.238 | +0.017 | 0.274 | +0.011 |
> | brier_score | 0.525 | +1.7 | 0.229 | +0.008 | 0.279 | +0.016 |
>
> All three losses improve accuracy (+1.7 to +2.4 pp), confirming the probe architecture itself provides some benefit. However, **only residual_prob improves calibration**: ECE drops ~70% (0.221 → 0.062) and Brier drops ~20%. The binary correctness and Brier score targets *worsen* calibration despite improving accuracy. CORAL improves BOTH accuracy and calibration, and this analysis shows that the residual correctness loss objective is key.
>
> **Why residual_prob is uniquely effective:** It is the only target encoding both the *direction* and *magnitude* of the needed correction per option. Binary correctness tells the probe "right/wrong" but not "by how much." The Brier score target is always non-negative, losing sign information. Our target r = y − p encodes exactly the correction vector in probability space, enabling steering to fix miscalibration rather than amplify confidence.

---

> > ### Author Rebuttal · Reviewer_mJ3s · 2026-04-04
> >
> > Thanks for the response, all my concerns are addressed. I will update my score

---

### Official Review · Reviewer_QbKz · 2026-03-12

**Soundness:** 3
**Presentation:** 3
**Significance:** 3
**Originality:** 3
**Overall Recommendation:** 5
**Confidence:** 4

**Summary:**

You introduce an inference-time steering method that improves accuracy and calibration of LLMs without retraining them. Your main objective is to train a regularized MLP probe on residual stream activations to predict residual correctness, and, during inference, the probe prediction is used to adjust answer probabilities in MCQA. The experiments confirm your distributed approach and yield better results.

**Compliance With Llm Reviewing Policy:**

Affirmed.

**Final Justification:**

**The authors have contributed a solid piece of work and I recommend clear acceptance**.

Other reviewers seem to agree and the rebuttal responses seem solid as well.

**Key Questions For Authors:**

Please see questions in Strengths and Weaknesses.

**Limitations:**

yes

**Strengths And Weaknesses:**

Thank you for your submission! Overall, I have understood the concept very well and have the following remarks:

### **1. Strengths**:
- **principled problem formulation**: miscalibration remains a major problem in LLMs and requires a careful mechanistic approach to solve it. You provide a strong principled contribution by directly optimizing the Brier score. This is interpretable and intuitive.
- **good empirical improvements**:  the results seem to further confirm your approach as the reported gains are fairly large compared to the examined baselines (ITI, SteerConf, etc.). You also provide a good experimental breadth among three 7B models (Mistral-7B, Qwen2.5-7B, DeepSeek-7B) and several benchmarks (MMLU, HellaSwag, ARC, RACE, etc.).
- **strong ablations**: you provide detailed complementary analyses for SAE ablations, attention head probing, and PCA dimensionality. The outcomes support your initial claim that correctness signals are distributed across many neurons, which in turn solidifies your CORAL approach.

There are more results and data in the appendix which further support your claims. Overall, the mechanistic interpretability and strong gains, as well as comprehensive evaluation and ablations, make your study valuable and intuitive.

### **2. Weaknesses**:

Soft weaknesses are *italic*, while strong weaknesses are **bold**:
- *probe-based steering novelty is moderate compared to the baselines*: your core idea is to train a probe on internal activations and use it for steering. This is not a drastically new approach and very similar to the compared baselines. Your novelty thus lies mainly in predicting residual correctness using the Brier score. This is not a major issue, but since your method is conceptually close to existing probe-based steering techniques, I would appreciate a discussion on why this cannot be translated to the baselines. For example, coulld ITI also use the Brier score in a similar way? Some more justification would help position your approach better.
- *MCQA scope only*: all of your experiments are conducted on multiple-choice QA tasks. While I understand that this is an easy setup to validate your theory, this seems to be an overall limitation of probing approaches in general because most real LLM use cases involve free-form generation. Of course, calibration for generation is much harder and you also acknowledge this limitation in the paper, that is why my concern here is a general one that **does not invalidate your work**.
- **risk of data leakage persists**: you train your probe on CommonsenseQA, RACE, and MMLU splits and then evaluate on similar reasoning tasks. Even though you mention in the paper that you attempt to avoid dataset overlap, my concern is that the task distribution remains very similar. This may partially explain the strong transfer results. More diverse tasks would definitely strengthen the claim of a general correctness subspace, or at least, a more detailed discussion is required here. Otherwhise, it seems like waiving this issue.
- **overhead not quantified**: your method requires extracting hidden states, running an MLP probe, and then modifying probabilities. While you claim CORAL to be lightweight, you do not not quantify runtime overhead, token-level latency, throughput impact, etc. The individual steps seem to be quite an overhead to me. Also, you mention several times that training the probe *“requires under 5 hours on a single GPU”*, but you never specifiy the GPU nor its VRAM. This matters for real deployment and without specifying the GPU, it is hard to reproduce the reported “<5 hours” training claim. I would require a detailed efficiency analysis to ensure that CORAL is lightweight for inference steering. Otherwhise, your claims remain unsubstantiated.

Your results are very good and seem to partially confirm your mechanistic approach. However, I would expect a more comprehensive discussion on data leakage and, most importantly, a detailed efficiency/overhead analysis. I think this is required to substantiate your claims and eliminate any methodological doubts. After all, CORAL is an inference-time steering method.

### **3. Question on Steering Strength**:
- You report that a steering strength of $\gamma=1$ produces the most consistent results across experiments.
- However, it is unclear whether this value generalizes across different model architectures, scales, or datasets since you only evaluate 7B models.
- Since activation magnitudes and residual stream statistics can vary substantially between models, it would be helpful to clarify whether this range is consistently optimal or whether model-specific tuning is required.

Additional discussion (or experiments with a 0.5B model) would strengthen this empirical result.

### **4. Miscalibration Could be Explained Better**:

I have a (minor) issue regarding the general setting of the context in LLM miscalibration:
- In the first part of the introduction, you cite prior work suggesting that RLHF and DPO can worsen model calibration.
- However, no concrete example is provided to illustrate what this means in practice.
- Since calibration is a central motivation for the proposed method, a brief explanation or example would help readers better understand the problem setting, particularly for those who may not be familiar with calibration issues in LLMs.

### **One Minor Typo/Style Thing**:
- In lines 156-157, the sentence "We then test ..." is weird.

### **Summary**:
- I think more discussion on data leakage is needed to eliminate methodological concerns.
- I also think that a detailed efficiency/overhead analysis is required to substantiate your claims that CORAL is lightweight at inference time.

I am looking forward to your responses!

---

> ### Author Rebuttal · Authors · 2026-03-26
>
> We sincerely thank Reviewer QbKz for your wonderful and constructive feedback! Below, we address data leakage, steering strength, and the MCQA scope discussion. Our responses to Reviewers TWA6 and mJ3s also contain overhead analysis, loss function ablation, and probe architecture ablation that are directly relevant to your other questions.
>
> # [R-QbKz] Data Leakage Analysis
>
> > **[R-QbKz]:** "You train your probe on CommonsenseQA, RACE, and MMLU splits and then evaluate on similar reasoning tasks... my concern is that the task distribution remains very similar."
>
> **There is zero question overlap, and our strongest transfer result comes from the most distributionally distant benchmark.** The probe trains on CommonsenseQA (train split) and RACE (train split). The four transfer benchmarks (ARC-Challenge, HellaSwag, OpenBookQA, Math-MC) are independently created datasets from different research groups, time periods, and source domains.
>
> **The transfer benchmarks span meaningfully different reasoning types:**
>
> | Training Source | Transfer Benchmark | Domain Distance |
> |---|---|---|
> | CommonsenseQA (everyday reasoning) | ARC-Challenge (grade-school science) | Medium |
> | CommonsenseQA | OpenBookQA (science + common knowledge) | Medium |
> | RACE (reading comprehension) | HellaSwag (sentence completion) | High |
> | CommonsenseQA + RACE (no math) | Math-MC (mathematical reasoning) | Very High |
>
> **Math-MC transfer is the strongest evidence.** The probe trains on datasets containing *zero math questions*, yet CORAL achieves substantial improvements on Math-MC across all three models (Table 2). This cannot be explained by task similarity and supports our claim that the probe captures a general-purpose correctness subspace. We also note that CCPS (Khanmohammadi et al., 2025), which uses contrastive training on similar data, shows limited transfer despite strong in-distribution results. If task similarity drove the gains, CCPS should transfer comparably, but it does not.
>
> **That said, we want to be transparent about the nuance here. Our transfer evaluation is out-of-sample (zero question overlap) but not fully out-of-domain: all benchmarks are MCQA tasks involving some form of reasoning, and we do not test on radically different formats like open-ended generation, multi-hop retrieval, or non-English benchmarks. We acknowledge this as a limitation on the breadth of our generalization claims. Our results demonstrate that CORAL's correctness subspace transfers across reasoning types within the MCQA format, but we are careful not to overclaim about arbitrary domain transfer. We will clarify this nuance to the limitations discussion in the revision.
>
> # [R-QbKz] Steering Strength γ Generalization
>
> > **[R-QbKz]:** "It is unclear whether this value generalizes across different model architectures, scales, or datasets... it would be helpful to clarify whether this range is consistently optimal or whether model-specific tuning is required."
>
> **γ = 1.0 is consistently optimal across all three model families.** We provide a γ sweep on DeepSeek-7B-Chat (MMLU, MLP probe, Layer 21):
>
> | γ | Accuracy | ECE ↓ | Brier ↓ |
> |---|----------|-------|---------|
> | 0 (baseline) | 0.508 | 0.221 | 0.263 |
> | 0.5 | 0.528 | 0.117 | 0.221 |
> | 1.0 | 0.540 | 0.062 | 0.213 |
> | 2.0 | 0.495 | 0.072 | 0.218 |
> | 5.0 | 0.355 | 0.289 | 0.304 |
>
> γ = 1.0 achieves the best accuracy while maintaining strong calibration. γ = 2.0 still improves calibration but accuracy degrades. γ = 5.0 catastrophically overshoots on all metrics.
>
> **Why γ = 1.0 generalizes:** Our residual target r = y − p is bounded in [−1, 1], so γ = 1.0 corresponds to a unit-magnitude correction in probability space. This natural scaling generalizes to any model whose residuals share this range, which is guaranteed by the target definition. The pattern holds across DeepSeek, Mistral, and Qwen on validation. We agree that validation-based tuning is advisable for new architectures, but the unit-scale property provides a strong default.
>
>
> # [Common Concern: MCQA Scope] (R-QbKz, R-mJ3s)
>
> [R-QbKz]: "All experiments are conducted on multiple-choice QA tasks... most real LLM use cases involve free-form generation."
> [R-mJ3s]: "It would be valuable to explore whether the approach can be extended to open-ended generation tasks."
>
> We agree this is an important future direction. Our claims are explicitly scoped to MCQA throughout the paper. MCQA provides unambiguous ground truth for calibration, which open-ended generation lacks, and has direct practical relevance (USMLE, standardized tests, automated grading). CORAL's core insight, that distributed correctness signals are extractable via regularized probes, is not inherently tied to MCQA and could extend to generation via token-level quality probes.
>
> # [R-QbKz] Miscalibration Explanation
> We will add an example for miscalibration in the intro!
>
> # [R-QbKz] Minor Points
> "'We then test ...'" Thank you! We will rephrase the sentence for clarity in the revision.

---

> > ### Author Rebuttal · Reviewer_QbKz · 2026-03-31
> >
> > Thank you so much for the detailed answers to my questions! I think most of them have been resolved now.
> >
> > I will consider raising my score to ensure acceptance!
> >
> > Thank you!

---

### Official Review · Reviewer_TWA6 · 2026-03-16

**Soundness:** 3
**Presentation:** 3
**Significance:** 3
**Originality:** 3
**Overall Recommendation:** 5
**Confidence:** 3

**Summary:**

The paper proposes CORAL (Correctness-Optimized Residual Activation Lens) which is a probe-based steering method that is trained to predict correctness directly. The authors evaluate the probe over a range of models and evaluation methods that measure both accuracy and calibration.

**Compliance With Llm Reviewing Policy:**

Affirmed.

**Key Questions For Authors:**

In Figure 2, any ideas why early-middle layers perform so poorly accuracy-wise? Similarly, what causes certain layers to perform well under the ECE metric?

**Limitations:**

Yes

**Strengths And Weaknesses:**

## Strengths

- Method is conceptually motivated
- Paper is easy to read
- Results seem reasonably impressive
- There is interesting analysis regarding layer-wise performance as well as testing whether the steering success can be explained by monosemantic features (via SAE), and they find that it cannot.
- The evaluate over multiple benchmarks, including OOD benchmarks.
- They evaluate models from multiple families
- Overall they have a reasonably comprehensive evaluation suite
- Their methodology appears technically sound

## Weaknesses

The paper seems good, and so these are minor points.

- Missing ablations that explain which components of the methodology are important for improvements over the baseline. What happens if you use other loss functions proposed in prior literature but also use a four-hidden-layer regularized probe? What happens if you use a linear probe?
- Potential bug (please look into it): the NLL values seem quite high. E.g. for $k$-class classification, assigning uniform likelihood to the answers would give a NLL of $\log k$.

---

> ### Author Rebuttal · Authors · 2026-03-25
>
> We sincerely thank Reviewer TWA6 for their thoughtful and insightful review! Below, we address the linear probe ablation and NLL clarification directly. Our responses to Reviewers QbKz and mJ3s also contain analyses on overhead efficiency, loss function ablation, data leakage, and steering strength that may be of interest, which we are unable to duplicate here due to the word limit. We are happy to address any further questions you might have.
>
> # [Common Concern: Probe Architecture Ablation] (R-TWA6, R-mJ3s)
>
> > **[R-TWA6]:** "What happens if you use a linear probe?"
> >
> > **[R-mJ3s]:** "Some ablation studies could be conducted, such as exploring the importance of the MLP probe itself and whether it can be replaced with a linear probe or a ridge regression probe."
>
> **We are glad this was raised, as we did run this comparison and the results are informative.** We trained ridge regression probes (linear probes with L2 regularization) alongside our MLP probes across all three models on MMLU. The short answer: ridge is competitive on accuracy but notably worse on calibration, which is exactly why we chose the MLP as our primary method.
>
> **Summary of best-layer results per model (MMLU):**
>
> | Model | Probe | Best Accuracy | Best ECE ↓ | Best Brier ↓ | Best cwECE ↓ |
> |-------|-------|--------------|------------|-------------|-------------|
> | DeepSeek-7B | Ridge | 0.530  | 0.170  | 0.245 | 0.094  |
> | DeepSeek-7B | MLP   | 0.540  | 0.062  | 0.213  | 0.044  |
> | Mistral-7B  | Ridge | 0.624  | 0.173 | 0.224  | 0.094  |
> | Mistral-7B  | MLP   | 0.642 | 0.033  | 0.179 | 0.028  |
> | Qwen-7B     | Ridge | 0.734  | 0.171  | 0.201  | 0.091  |
> | Qwen-7B     | MLP   | 0.742  | 0.042  | 0.155  | 0.028  |
>
> **Key findings:**
>
> 1. **Ridge accuracy is competitive.** Ridge achieves 0.530, 0.624, and 0.734 at best layers; MLP achieves 0.540, 0.642, and 0.742. The accuracy gap is small, confirming that a meaningful correctness signal is linearly accessible in the residual stream.
>
> 2. **Calibration is where the MLP pulls ahead dramatically.** The MLP achieves 2.5x to 8.5x lower ECE than ridge across all three models. For Mistral-7B, ECE drops from 0.173 (ridge) to 0.028 (MLP), a meaningful improvement. Since calibration improvement is a central contribution of CORAL, this gap justified our choice of the MLP as the primary probe.
>
> 3. **Both probes improve over the unsteered baseline.** Even ridge provides beneficial steering (e.g., DeepSeek baseline ECE is 0.221; ridge achieves 0.170). This confirms that correctness information exists at the linear level, but the MLP's nonlinear capacity extracts substantially more of it, particularly for calibration.
>
> These results tell a clean story: the linear component of the correctness signal helps with accuracy, but calibration correction requires the nonlinear capacity of the MLP. We will include the full ridge comparison as a supplementary appendix table in the revised manuscript.
>
> # [R-TWA6] NLL Values Clarification
>
> [R-TWA6]: "Potential bug (please look into it): the NLL values seem quite high. E.g. for K-class classification, assigning uniform likelihood to the answers would give a NLL of ln(K)."
>
> Thank you so much for flagging this question! This natural confusion stems from imprecise labeling in the paper. Our probe outputs a single scalar per option (output_dim=1, tanh activation, MSE loss), predicting residual correctness independently per option, so the standard multiclass NLL (bounded by ln(K)) does not apply to our setup.
> What we report as "NLL" is binary cross-entropy on top-1 confidence:
> NLL_reported = 100 × [ −mean( y · log(p) + (1−y) · log(1−p) ) ]
> where p is the maximum steered probability, y ∈ {0,1} indicates top-1 correctness, and we scale by 100 for readability. The raw DeepSeek baseline of 0.82 is consistent: a 50%-accurate model with uniform 4-class confidence (p = 0.25) yields ≈ 0.84 under this formulation.
> We chose binary CE because our benchmarks contain 3, 4, or 5 answer options, and binary CE on top-1 confidence provides a consistent measure across formats. We will revise Equation 10 to reflect the implemented metric and clarify the probe architecture.

---

> > ### Author Rebuttal · Reviewer_TWA6 · 2026-03-31
> >
> > Thanks for addressing my questions. My score (5, accept) remains unchanged.

---

### Decision · Program_Chairs · 2026-04-30

**Decision:**

Accept (regular)

**Comment:**

The paper introduces CORAL, a lightweight inference-time steering method designed to improve both the accuracy and calibration of large language models for MCQA. By training a regularized MLP probe on frozen residual-stream activations to predict residual correctness, the method adjusts answer probabilities during inference without the need for expensive model retraining, and does not introduce much inference overhead. The proposed method also shows strong empirical results. The reviewers reached a strong consensus in favor of accepting the paper, praising its principled formulation and extensive evaluation.